# Understanding Task Transfer
# in Vision-Language Models

**Bhuvan Sachdeva**\*    **Karan Uppal**\*    **Abhinav Java**\*    **Vineeth N. Balasubramanian**
Microsoft Research India
Bengaluru, India

## Abstract

Vision–Language Models (VLMs) have achieved strong performance across diverse multimodal benchmarks through multitask training. Yet, these models struggle on visual perception tasks, falling way behind human-level performance. These models are often finetuned on a multitude of tasks. However, it is unclear how finetuning on one perception task influences zero-shot performance on others, a question that is crucial for designing efficient training strategies. In this work, we study how finetuning on one perception task affects the model's performance on other perception tasks and present the first systematic study of task transferability in VLMs within the perception domain. We introduce Performance Gap Factor (PGF), a novel metric that quantifies transfer by jointly capturing its breadth (how many tasks are affected) and magnitude (the strength of influence). Using three open-weight VLMs across 13 perception tasks, we construct a task graph that uncovers inter-task relationships previously unexplored in the multimodal setting. Our analysis reveals distinct cliques of mutually beneficial as well as mutually detrimental tasks. We also categorise tasks into different personas based on their transfer properties. These findings highlight both opportunities for positive transfer and risks of negative interference, offering actionable guidance for curating finetuning strategies and advancing general-purpose VLMs.

## 1 Introduction

Vision Language Models (VLMs) [Liu et al., 2023b, 2024a, Li et al., 2024a, Achiam et al., 2023, Li et al., 2024b, Wang et al., 2024] have demonstrated remarkable success across a wide range of multimodal benchmarks, including MMMU [Yue et al., 2024], DocVQA [Mathew et al., 2021b], InfoVQA [Mathew et al., 2021a], TextVQA [Singh et al., 2019], etc. Recent advances in VLMs have been driven largely by finetuning [Liu et al., 2023b], where a pretrained Large Language Model (LLM) is paired with a pretrained visual encoder and finetuned on curated multimodal datasets. These VLMs are then empirically studied on a wide range of tasks spanning diverse domains like image captioning, VQA, OCR, and interleaved image–text corpora, allowing them to acquire broad visual understanding capabilities. Despite recent varied efforts, there have been limited studies on understanding the connections between these different tasks, and how finetuning on one task can affect the VLM's performance on other tasks (a la Taskonomy [Zamir et al., 2018]). Complementarily, it is understood that VLMs lag behind human performance and specialist models on perception tasks such as depth estimation, object detection, and counting. On the widely followed BLINK benchmark [Fu et al., 2024], for instance, GPT4-V only achieves an overall accuracy of 51.14% across perception tasks, far below the human performance of over 95.67% [Fu et al., 2024]. We hence seek to address a key question in this work: *how finetuning on one task affects the zero-shot performance on other tasks*, with a focus on perception tasks for a focused and comprehensive analysis.

---

\*Equal contribution. Corresponding Author: `vineeth.nb@microsoft.com`

To this end, we define transferability as the degree to which finetuning on one task affects performance on others, encompassing both the *breadth* (how many tasks are influenced) and the *magnitude* (the average strength of that influence). We quantify this performance using a novel metric, *Perfection Gap Factor*, which we use to measure the transfer between any two tasks. We perform comprehensive analysis using three well-known VLMs of different model sizes on diverse perception tasks, revealing critical insights into inter-task relationships and impact of task granularity. We show how different perception tasks interact and identify properties that may help predict positive versus negative transfer.

Prior methods have explored task correlations through transfer learning Chen et al. [2024], Zamir et al. [2018], which differs from our setting as we do not finetune on the target task. DEFT [Ivison et al., 2022] leverages nearest neighbors to determine data mixtures for text models for efficient finetuning. Another work [Tiong et al., 2024] performs large-scale transfer learning experiments on VLMs through the lens of evaluation and proposes a new benchmark. LLaVA-OneVision [Li et al., 2024a] proposes a new model family that allows enhanced transfer between different modalities like video and images. In contrast, to the best of our knowlege, ours is the first work that studies the transferability between perception tasks in the context of VLMs.

**Summary of Contributions.** We present the first systematic study of perception task-transfer in VLMs, evaluating state-of-the-art open-weight models across 13 diverse perception tasks, establishing a transfer map that was previously unexplored beyond CNN-based vision models. *Second*, we formally define task transferability considering two key axes – (a) *how many tasks are influenced?*, (b) *what is the average magnitude of influence?*. *Third*, we propose a novel metric – **Perfection Gap Factor** (PGF), which quantifies the degree of transferability between any two tasks. *Finally*, we present our analysis providing practical insights about finetuning behaviors, highlighting key properties such as cliques of mutually beneficial tasks, scale-dependent task transfer trends, and distinct task roles. Beyond understanding task generalization capabilities of VLMs, our inferences provide actionable guidance for designing more effective training regimes for VLMs.

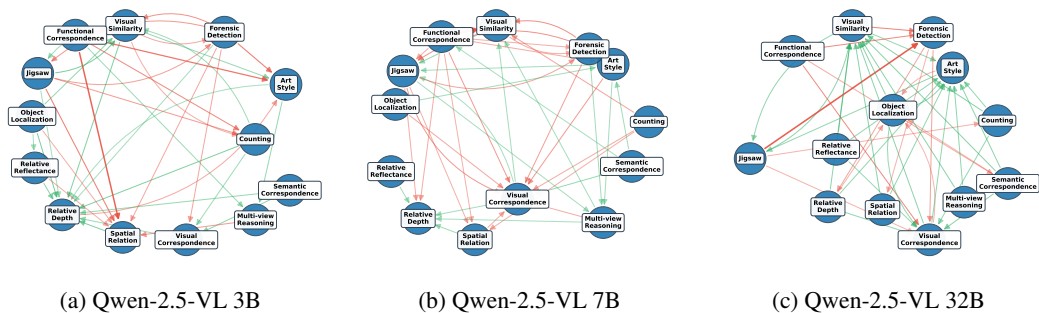

(a) Qwen-2.5-VL 3B        (b) Qwen-2.5-VL 7B        (c) Qwen-2.5-VL 32B

Figure 1: Task transfer graphs for different sizes of Qwen-2.5-VL.

## 2 Methodology

We herein present our framework for characterizing the behavior of VLMs across diverse perception tasks. We begin by defining task transferability, which measures the extent to which finetuning to one task alters performance on others. We then quantify transferability using our proposed metric **Perfection Gap Factor (PGF)**, which provides a robust lens on cross-task VLM malleability.

**Preliminaries.** We consider the setting where a VLM $\mathcal{M}$ is fine-tuned on a source task $T_{\text{S}}$ using a source dataset $\mathcal{D}_{\text{S}}^{\text{train}}$ and subsequently evaluated on a set of $N$ target tasks $\{T_j\}_{j=1}^N$ using target datasets $\{\mathcal{D}_j^{\text{eval}}\}_{j=1}^N$. The central question we study is on how fine-tuning on $T_S$ affects performance on $\{T_j\}_{j=1}^N$, and how one can measure such inter-task relationships. We denote the finetuned model on a given dataset as $\mathcal{M}(\cdot)$. Throughout the paper, fine-tuning refers exclusively to use of LoRA Hu et al. [2022] on the training split of $T_{\text{S}}$, while evaluation is performed using the evaluation split of the corresponding task. We define this relationship between tasks formally below.

**Definition 1 (Task Transferability)** *Let $\mu_{i \to j}$ denote the transferability score from source task $T_i$ to target task $T_j$, $p = |\{j : \mu_{i \to j} > 0\}$ as the number of positive scores, and $n = |\{j : \mu_{i \to j} < 0\}|$ as the number of negatives scores. The **positive** and **negative** task transferability of a source task $T_i$*

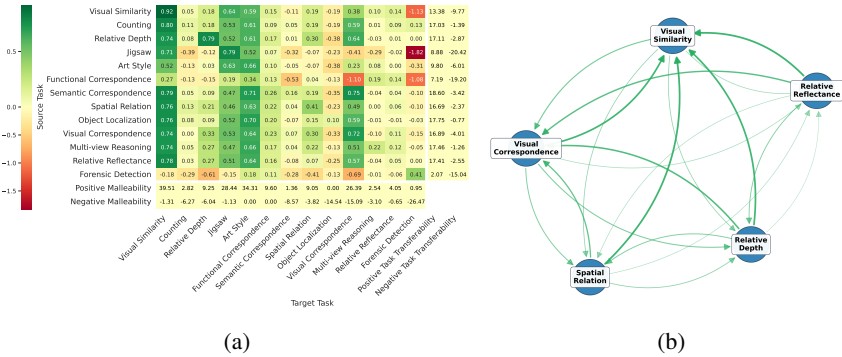

|  | (a) | | | | | | | | | (b) |

Figure 2: (a) Perfection Gap Factor Heatmap for Qwen-2.5-VL 32B. (b) Positive clique found in Qwen-2.5-VL 32B analysis.

*to a set of target tasks $\{T_j\}_{j=1}^{N}$ is given by,*

$$\Delta(i)^{+} := \left(\frac{1-e^{-\frac{p}{N}}}{p}\right)\sum_{j=1}^{N}\mu_{i \to j}\,\mathbf{1}_{\{\mu_{i \to j} > 0\}}, \quad \Delta(i)^{-} := \left(\frac{1-e^{-\frac{n}{N}}}{n}\right)\sum_{j=1}^{N}\mu_{i \to j}\,\mathbf{1}_{\{\mu_{i \to j} < 0\}} \quad (1)$$

*where $\Delta(i)^{+}$ refers to positive task transferability and $\Delta(i)^{-}$ refers to negative task transferability.*

**Perfection Gap Factor.** We define the **PGF** between $T_i$ and $T_j$ as the *ratio of performance gain to the perfection gap*, i.e.,

$$\mu_{i \to j} = \frac{\mathrm{Acc}(\mathcal{M}(T_i),\,T_j) - \mathrm{Acc}(\mathcal{M},\,T_j)}{U_j - \mathrm{Acc}(\mathcal{M},\,T_j) + \epsilon} \quad (2)$$

In Eq 1, we consider the average positive and negative **PGF** along with the spread of influence using the logistic-weighted average over tasks. Finally, our metric, **PGF** is bounded by a finite positive and negative value – 1 and by a function of the upper bound performance $U_j$ when the transferability is positive and negative respectively. Since the upper and lower bounds of **PGF** are not symmetric, we compute the positive and negative averages independently.

## 3 Results and Analysis

**Setup.** We consider a diverse set of 13 multimodal perception tasks, from the widely followed BLINK Benchmark Fu et al. [2024]. A detailed description of these tasks can be found in the Table 1. As base models, we use three open-weight variants of Qwen-2.5-VL Wang et al. [2024] (3B, 7B, 32B), which we finetune with LoRA Hu et al. [2022]. Finetuning is performed independently on each task, and evaluation is carried out on the validation splits of all tasks. Since BLINK itself only provides validation and test splits, we reconstruct training data by retrieving the original datasets used in BLINK, adhering to the same task definitions and response formats (see Appendix C for details). We exclude the "IQ Test" task from our analysis because it was manually constructed and does not have a corresponding training set.

### 3.1 Cliques of Cooperation

The improvements across tasks are not uniformly distributed and instead, exhibit structured clusters of mutual benefit. We define such a cluster as: Let $T$ be the set of tasks. A subset of tasks $C \subseteq T$ forms a clique if, for every task $T_i \in C$, finetuning on $T_i$ induces $\mu_{i \to j}$ of consistent sign (all negative or all positive) with all of the remaining tasks in $C$. We refer to cliques with consistently positive PGF as positive cliques and consistently negative PGF as negative cliques. Since Qwen-2.5-VL 32B exhibits the strongest task transferability (both positive and negative) across all tasks, we focus our clique analysis on this model.

We identify 9 maximal positive cliques amd 2 maximal negative cliques. An example for the same is illustrated in Figure 2b. Examining Qwen-2.5-VL 3B and 7B reveals similar cliques, although

they are less pronounced. We present detailed analysis in Appendix D.2. Moreover, surprisingly, we note that {Visual Correspondence, Semantic Correspondence, Functional Correspondence} do not form a clique, even though their tasks are quite inter-related.

## 3.2 Bigger Models, Better Transfer

We plot the task transfer graphs in Figure 1. To understand how task transferability varies with increasing model size, in Figure 3, we analyze the average positive and negative task transferability across all tasks for each model. As expected, as model size increases, the average positive transferability also increases. This finding aligns with the intuition that increased representational capacity allows models to capture more generalizable features, leading to better transfer of beneficial knowledge across diverse tasks. However, there is no consistent trend with the average negative transferability. We provide transferability heatmaps for 3B and 7B models in the Appendix D.1.

## 3.3 Task Personas: Donors, Pirates, Sponges, and Sieves

To investigate how different tasks interact with each other in detail, we classify the source tasks on the basis of their transferability into *Donors and Pirates* and the target tasks on the basis of their malleability into *Sponges and Sieves*.

**Donors and Pirates**: We refer to tasks that consistently exhibit high positive transferability across models as *Donor Tasks*, and tasks that tend to induce negative transfer across models as *Pirate Tasks*. *Visual Similarity*, *Object Localization* and *Semantic Correspondence* emerge as *Donors*, while *Functional Correspondence* and *Jigsaw* behave as Pirates, as shown in Figure 3.

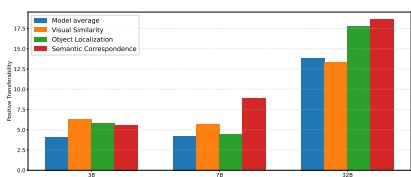

(a) Trend of positive task transferability.

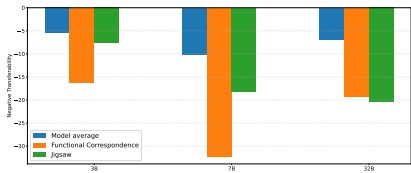

(b) Trend of negative task transferability.

Figure 3: Task transferability trends across model sizes of Qwen-2.5-VL.

**Sponges and Sieves**: A target task can be considered highly *malleable* if finetuning on different tasks leads to significant change in performance on that task. The sign of this change determines whether the task is *positively malleable* or *negatively malleable*. To capture the *positive* and *negative malleability* of a target task, we compute the average of the positive and negative PGF scores induced on that task by other source tasks, scaled with Logistic Weighting (as in Transferability). This weighting emphasizes tasks that benefit from many others, and conversely highlights tasks that are consistently hindered. In this sense, we view tasks with high Positive Malleability across models as *Sponge Tasks*, while those with high Negative Malleability as *Sieve Tasks*. *Relative Depth* demonstrates high positive malleability, acting as a *Sponge*, as shown in Figures 2a and Figure 5 in Appendix D.1. *Visual Correspondence* emerges as a *Sieve* showing deterioration in accuracy across tasks and models.

## 4 Conclusion

In this work, we present the first systematic analysis of perception task transfer in vision-language models. To facilitate this analysis, we introduce a new metric called Perfection Gap Factor, which helps us quantify perception task transfer in VLMs. Through experiments with three state-of-the-art VLMs, we study how finetuning on a source task impacts zero-shot performance on other tasks. Our analysis reveals several key insights. Firstly, we note that positive task transferability increases with model size. Secondly, we identify distinct cliques of mutually beneficial and mutually detrimental tasks. Lastly, we investigate how inter-task interactions and characterise them into task personas. This analysis provides actionable insights into how task interactions shape model behavior, guiding the development of finetuning strategies to enhance general-purpose VLMs.

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

# A  More Related Work

**Vision-Language Models (VLMs).** Vision–Language Models (VLMs) have significantly advanced in visual understanding, in part due to techniques such as Visual Instruction Tuning Liu et al. [2023c]. Subsequent work has further enhanced these models through scaling to high-resolution images using any-res Liu et al. [2024b] and windowed attention Fu et al. [2025] supporting high fidelity tasks such as fine-grained VQA and OCR. In addition, expanding training datasets to include a diverse range of tasks—particularly multi-image tasks Li et al. [2024a]—has been shown to improve performance on video tasks, demonstrating task transfer capabilities from multi-image to the video domain. Reasoning within VLMs has also been studied through Chain-of-Thought-style approaches, using bounding-box prediction or region selection as reasoning steps Wu and Xie [2024], Shao et al. [2024], Bigverdi et al. [2025].

**BLINK Benchmark.** The BLINK Benchmark Fu et al. [2024] focuses on the visual perception capabilities of VLMs. The benchmark comprises 14 tasks and each task contains multiple choice questions with single or multiple (2-5) images per question. A detailed list can be found in Table 1.

**Task Transfer.** The Taskonomy framework Zamir et al. [2018] investigates task transferability across a wide range of computer vision problems (e.g., image classification, semantic segmentation, depth prediction, image inpainting). Their approach involves pretraining an encoder on a source task and then training a task-specific decoder on a target task, enabling estimation of transferability scores between tasks. Sundaram et al. [2024] show that finetuning vision backbone models on image similarity triplets (similar to preference tuning in LLMs) benefits a variety of downstream tasks, such as depth prediction, counting, image retrieval and segmentation. The authors argue that doing so aligns the model's latent representation with human preferences, thus leading to performance improvements. Huan et al. [2025] examine how finetuning models on mathematical reasoning tasks affects their performance on both general reasoning and non-reasoning tasks. They also introduce a task-transferability index—defined as the accuracy gain relative to baseline scores—to quantify these interactions.

# B  Behavior of Perfection Gap Factor (PGF)

Figure 4 illustrates how the Perfection Gap Factor (PGF) varies with baseline performance $x$ and accuracy change $k$ after finetuning. Several numerical properties emerge:

- **Positive Bound**: For improvements ($k > 0$), PGF is capped at 1, achieved when finetuning fully closes the gap to perfection ($k = 100 - x$).
- **Negative Bound**: For deterioration ($k < 0$), PGF admits a finite lower bound due to accuracy discreteness. With $m$ evaluation questions, the highest baseline strictly below 100% is $x = 100 \left(1 - \frac{1}{m}\right)$. The worst deterioration is $k = -x$ (accuracy drops to zero), yielding

$$\mathrm{PGF}_{\min} = \frac{-x}{100 - x} = \frac{-100(1 - \frac{1}{m})}{100/m} = -(m - 1).$$

  For instance, with $m = 200$ questions, $\mathrm{PGF}_{\min} = -199$. The worst-case deterioration therefore grows linearly with $m$.
- **Asymmetry**: Since positive PGF is capped at 1 but negative PGF can reach $-(m-1)$, PGF is inherently asymmetric, motivating our separate study of positive vs. negative transferability.
- **Ceiling Sensitivity**: Near-perfect baselines amplify PGF: small accuracy shifts yield disproportionately large values. This highlights ceiling-level improvements while penalizing degradations more harshly.

# C  Implementation Details

All training is performed on 8xA100s 40GB. DeepSpeed [Rasley et al., 2020] ZeRO-2 is used for Qwen-2.5-VL 3B and 7B, while DeepSpeed [Rasley et al., 2020] ZeRO-3 is used for Qwen-2.5-VL 32B, all with mixed-precision. Batch size is set to 16, weight decay as 0 and warmup ratio of 0.03 with cosine decay learning rate scheduler. For finetuning, LoRa rank is set to 8 for all tasks except

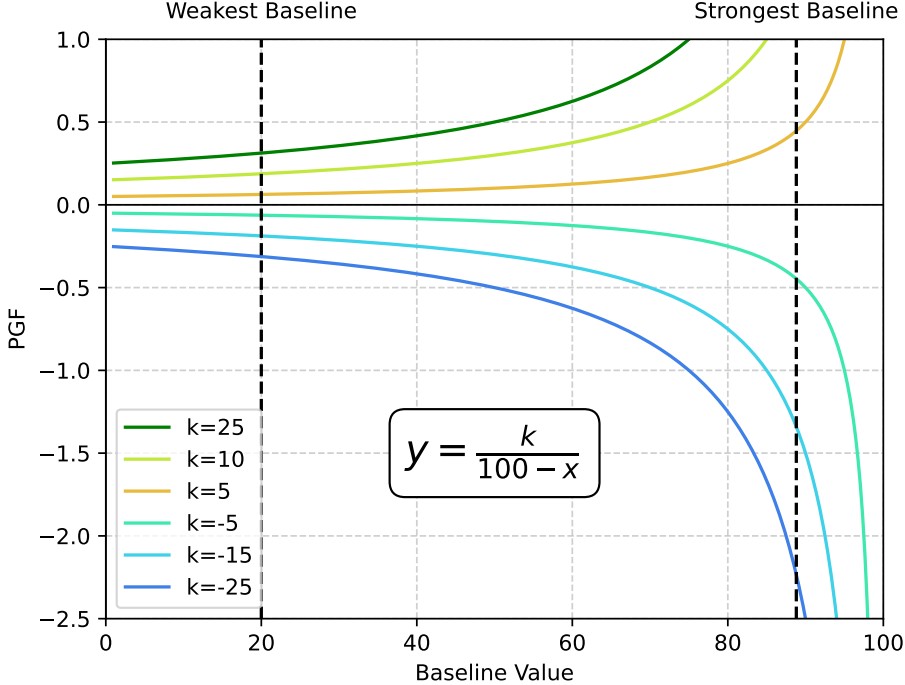

Figure 4: Behavior of PGF as a function of baseline accuracy ($x$) and change after finetuning ($k$).

Object Localization. For Object Localization, we use a rank of 64 to ensure convergence. $\alpha$ is set to 16 for all tasks. Task-wise training details are mentioned in Table 1.

# D   Additional Results

## D.1   Detailed Heatmaps for Qwen2.5-VL 7B and 3B

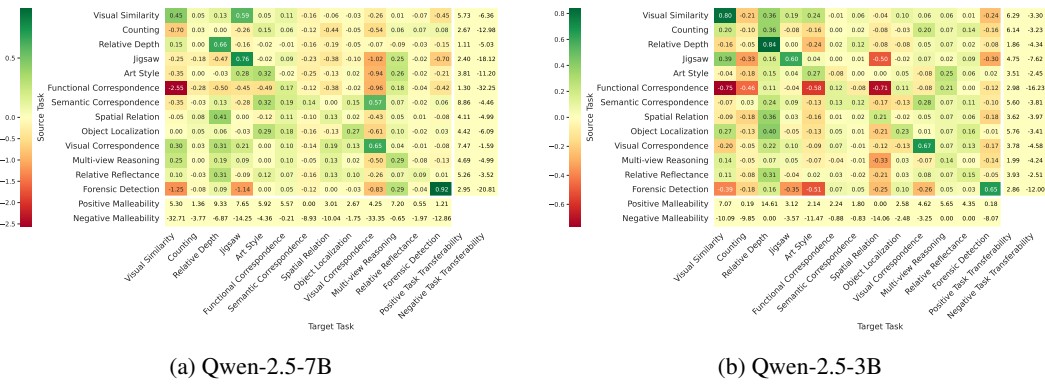

(a) Qwen-2.5-7B          (b) Qwen-2.5-3B

Figure 5: Task transferability analysis for smaller Qwen models.

## D.2   Detailed Clique Graphs

We present the 9 positive and 2 negative cliques found in Qwen-2.5-VL 32B, as well as examine similar trends in Qwen-2.5-VL 3B and 7B in Figure 2.

[2]https://huggingface.co/datasets/keremberke/painting-style-classification

| Task | Description | Source Dataset | Hyperparams |
|---|---|---|---|
| Visual Similarity | *Given a reference image alongside two alternatives, identify the image most visually similar to the reference.* | DreamSim (Nights) [Fu et al., 2023] | 15,914 examples, 10 epochs, 1e-3 lr |
| Counting | *Given an image, a counting-related question, and 4 options, choose the correct answer.* | TallyQA [Acharya et al., 2018] | 250k examples, 1 epoch, 1e-4 lr |
| Relative Depth | *Decide which of two specified points is closer.* | Depth in the Wild + Human Annotations [Chen et al., 2016] | 420k examples, 1 epoch, 1e-4 lr |
| Jigsaw | *Choose the image that completes the scene.* | TARA [Fu et al., 2022] | 11,837 examples, 5 epochs, 1e-3 lr |
| Art Style | *Given a reference painting and two candidate paintings, identify which shares the same art style.* | WikiArt[2] | 100k examples, 500 steps, 1e-3 lr |
| Functional Correspondence | *Match a reference point in one image with the best corresponding point among 4 options in another image, based on functional affordances.* | FunKPoint [Lai et al., 2021] | 100k examples, 1000 steps, 1e-3 lr |
| Semantic Correspondence | *Given a point in a reference image, choose the most semantically similar point among 4 options in another image.* | Spair-71k [Min et al., 2019] | 36k examples, 5 epochs, 1e-4 lr |
| Spatial Relation | *Identify the spatial relationship between objects in an image.* | Visual Spatial Reasoning [Liu et al., 2023a] | 7k examples, 5 epochs, 1e-4 lr |
| Object Localization | *Given an image and two bounding boxes (one ground-truth, one perturbed), choose the correct bounding box.* | LVIS [Gupta et al., 2019] | 18,912 examples, 10 epochs, 1e-4 lr |
| Visual Correspondence | *Identify the same point across two input images. One image has 1 point, the other has 4 candidate points.* | HPatches [Balntas et al., 2017] | 6k examples, 10 epochs, 1e-4 lr |
| Multi-view Reasoning | *Predict the direction of camera motion from two views.* | Wild 6D Fu and Wang [2022] | 4k examples, 10 epochs, 1e-4 lr |
| Relative Reflectance | *Decide which of two pixels is darker, or whether they have similar reflectance.* | Intrinsic Images in the Wild + Human Annotations [Bell et al., 2014] | 14k examples, 10 epochs, 1e-4 lr |
| Forensic Detection | *Identify synthetic images from a mixture of real and synthetic samples.* | Synthetic: COCO captions [Lin et al., 2015] + Stable Diffusion XL Real: COCO captions + Web search | 60,518 examples, 500 steps, 1e-3 lr |

Table 1: Overview of tasks used in our evaluation. Each task is paired with its source dataset and fine-tuning setup. The number of examples, epochs/steps, and lr are specified for each task.

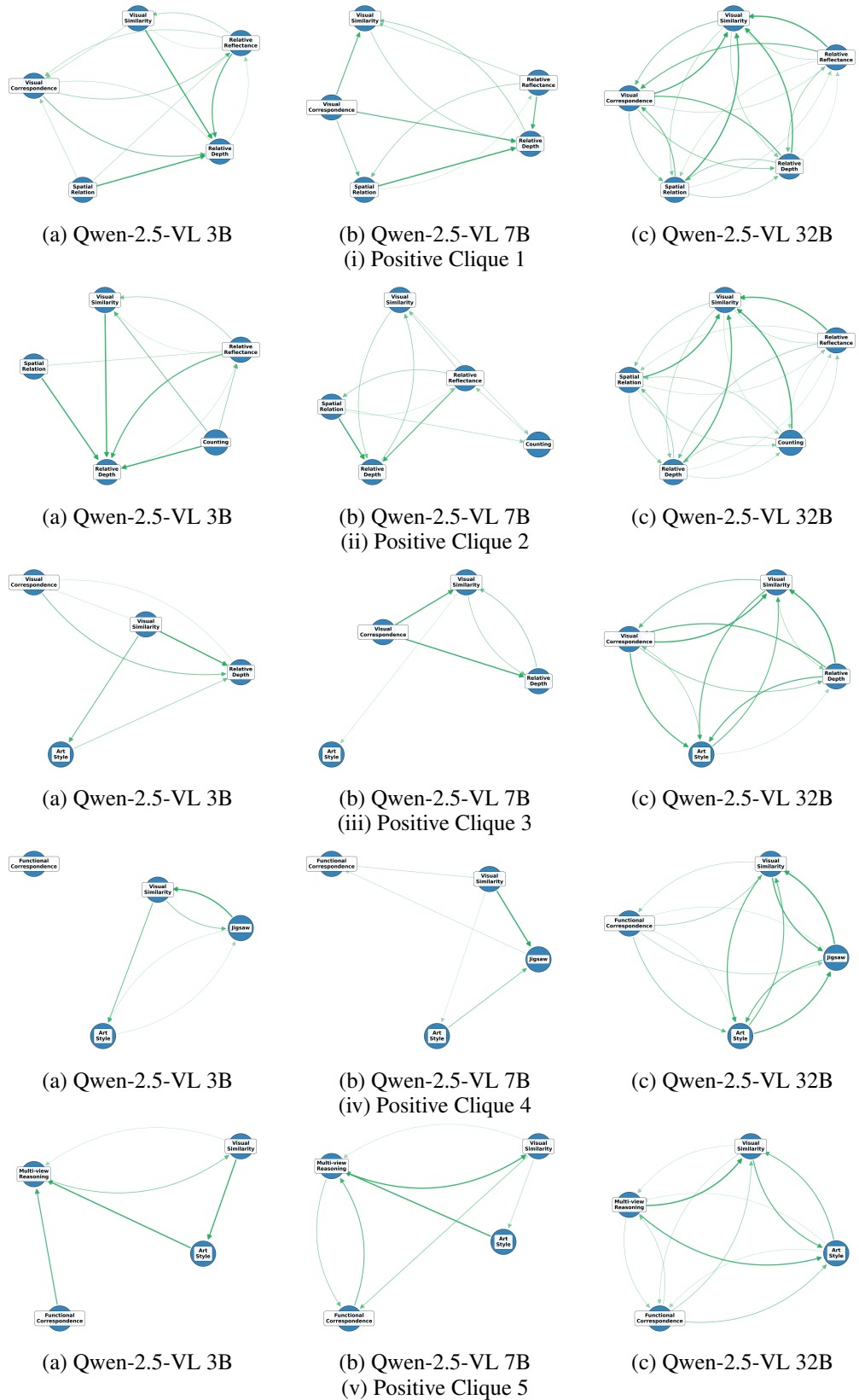

(a) Qwen-2.5-VL 3B     (b) Qwen-2.5-VL 7B     (c) Qwen-2.5-VL 32B

(i) Positive Clique 1

(a) Qwen-2.5-VL 3B     (b) Qwen-2.5-VL 7B     (c) Qwen-2.5-VL 32B

(ii) Positive Clique 2

(a) Qwen-2.5-VL 3B     (b) Qwen-2.5-VL 7B     (c) Qwen-2.5-VL 32B

(iii) Positive Clique 3

(a) Qwen-2.5-VL 3B     (b) Qwen-2.5-VL 7B     (c) Qwen-2.5-VL 32B

(iv) Positive Clique 4

(a) Qwen-2.5-VL 3B     (b) Qwen-2.5-VL 7B     (c) Qwen-2.5-VL 32B

(v) Positive Clique 5

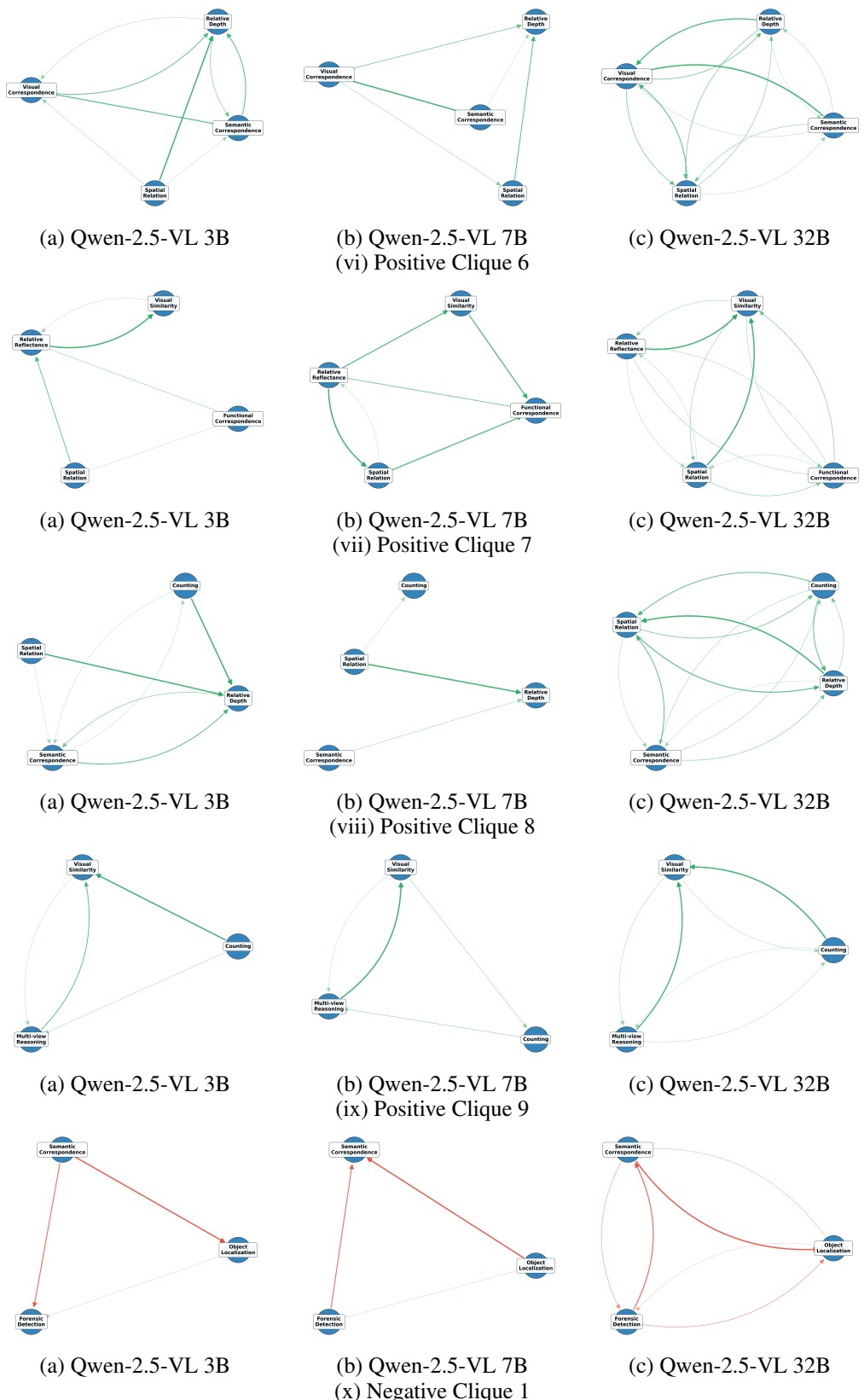

(a) Qwen-2.5-VL 3B   (b) Qwen-2.5-VL 7B   (c) Qwen-2.5-VL 32B
(vi) Positive Clique 6

(a) Qwen-2.5-VL 3B   (b) Qwen-2.5-VL 7B   (c) Qwen-2.5-VL 32B
(vii) Positive Clique 7

(a) Qwen-2.5-VL 3B   (b) Qwen-2.5-VL 7B   (c) Qwen-2.5-VL 32B
(viii) Positive Clique 8

(a) Qwen-2.5-VL 3B   (b) Qwen-2.5-VL 7B   (c) Qwen-2.5-VL 32B
(ix) Positive Clique 9

(a) Qwen-2.5-VL 3B   (b) Qwen-2.5-VL 7B   (c) Qwen-2.5-VL 32B
(x) Negative Clique 1

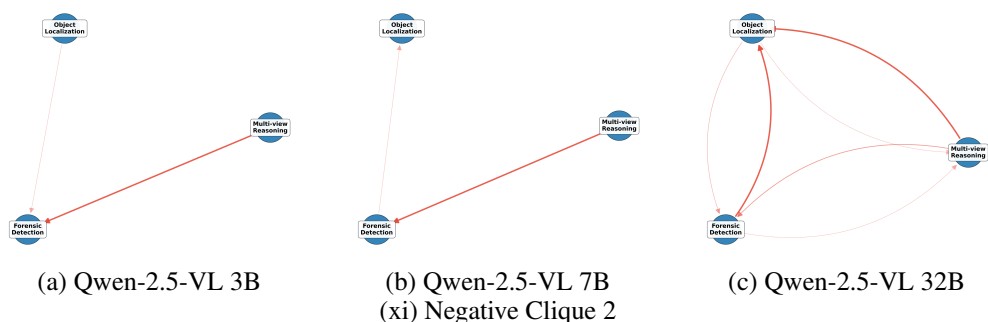

(a) Qwen-2.5-VL 3B     (b) Qwen-2.5-VL 7B     (c) Qwen-2.5-VL 32B

(xi) Negative Clique 2

Table 2: Cliques across all model sizes

## D.3 Category Wise Results

In this section, we analyze the positive and negative transferability across *task-categories*. We present our results on semantic categories (Pixel, Crop, Image Level Tasks) in Fig. 7 and on hierarchical categories (Low, Mid, High Level Tasks) in Fig. 6. Specifically, we refer to the BLINK [Fu et al., 2024] benchmark to categorize the tasks. The detailed category list is provided in Table 3. To compute category-wise task-transferability we compute the average positive and negative transferability of all tasks in a given category with all the tasks in the other category. Next, we define the category wise transfer formally.

Let $\mathcal{T}_a$ be the set of source tasks in category $a$ and $\mathcal{T}_b$ the set of target tasks in category $b$, with sizes $|\mathcal{T}_a|$ and $|\mathcal{T}_b|$. For each task $i \in \mathcal{T}_a$, let $p_{i \to b} := \sum_{j \in \mathcal{T}_b} \mathbf{1}_{\{\mu_{i \to j} > 0\}}$ denote the number of positive PGF scores induced by $\mathcal{T}_b$ after finetuning on $i$. Similarly, $n_{i \to b} := \sum_{j \in \mathcal{T}_b} \mathbf{1}_{\{\mu_{i \to j} < 0\}}$ denotes the number of negative PGF scores induced by $\mathcal{T}_b$ on task $i$. $N$ denotes the total number of target tasks. Then the positive and negative transferability from $a$ to $b$ are given by –

$$\Delta_{a \to b}^{+} := \frac{1}{|\mathcal{T}_a|} \sum_{i \in \mathcal{T}_a} \left[ \left( \frac{1 - e^{-\frac{p_{i \to b}}{N}}}{p_{i \to b}} \right) \sum_{j \in \mathcal{T}_b} \mu_{i \to j} \mathbf{1}_{\{\mu_{i \to j} > 0\}} \right] \tag{3}$$

$$\Delta_{a \to b}^{-} := \frac{1}{|\mathcal{T}_a|} \sum_{i \in \mathcal{T}_a} \left[ \left( \frac{1 - e^{-\frac{n_{i \to b}}{N}}}{n_{i \to b}} \right) \sum_{j \in \mathcal{T}_b} \mu_{i \to j} \mathbf{1}_{\{\mu_{i \to j} < 0\}} \right] \tag{4}$$

Intuitively, these quantities measure the magnitude of positive and negative transferability between two groups of tasks. Next, we discuss the key observations and analysis.

| Category | Tasks |
|---|---|
| Pixel | Relative Depth, Relative Reflectance, Visual Correspondence, Functional Correspondence, Semantic Correspondence |
| Crop | Jigsaw, Object Localization |
| Image | Visual Similarity, Forensics Detection, Counting, Art Style, Multi-view Reasoning, Spatial Relation |
| Low | Relative Depth, Relative Reflectance, Visual_Correspondence |
| Mid | Spatial Relation, Multi-view Reasoning, Jigsaw, Art Style |
| High | Functional Correspondence, Semantic Correspondence, Visual Similarity, Forensics Detection, Counting, Object Localization |

Table 3: Categories and their associated tasks.

**Semantic Task Transferability.** Fig. 7 illustrates the positive and negative transferability between semantic categories. On average, finetuning on pixel-level tasks yields the highest magnitude of both positive and negative transferability across all three models, with one exception: for Qwen2.5-VL-3B, the positive transferability of pixel-level tasks is a close second. This may be attributed to the fact that

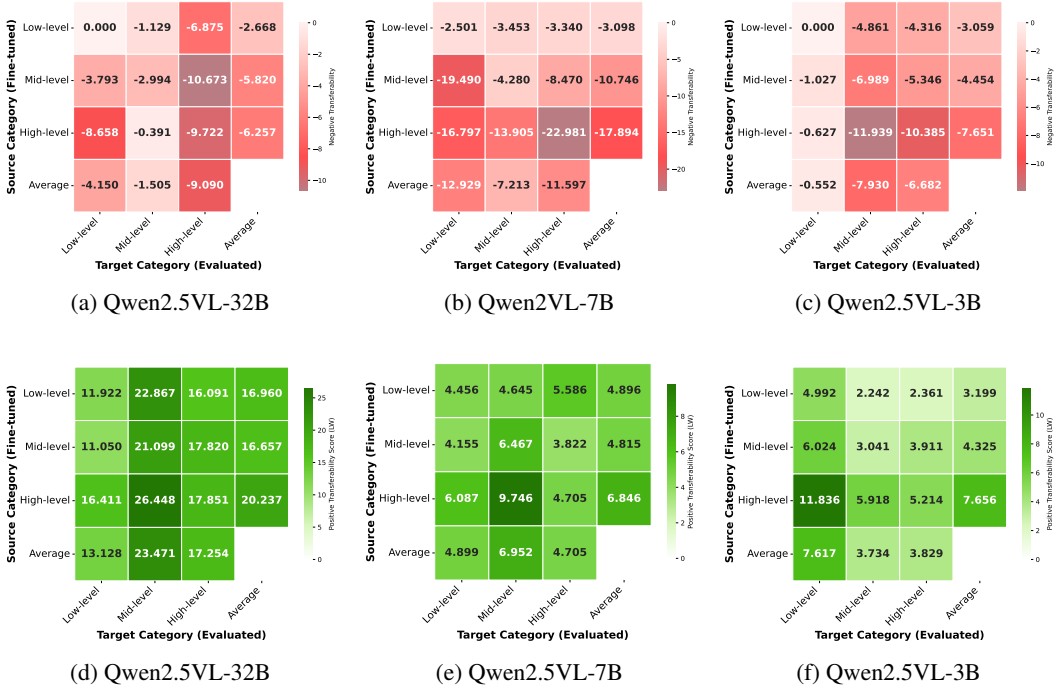

Figure 6: Comparison of transferability across hierarchical categories.

pixel-level tasks—often correspondence tasks—are generally harder to learn, and thus exert stronger influence on crop- and image-level tasks such as object localization and similarity. We further observe that, when finetuned on pixel-level tasks, the magnitude of positive transferability is consistently higher than that of negative transferability. Finally, for the 32B and 7B models, image-level tasks benefit the most on average from finetuning, achieving the highest positive transferability in two out of three cases.

**Hierarchical Task Transferability.** Fig. 6 illustrates the positive and negative transferability between hierarchical categories. We observe a consistent trend that finetuning on high-level tasks like functional correspondence, semantic correspondence, forensic detection, etc has the highest average magnitude of task transferability. This is similar to the trend observed in semantic task transferability and may be because high-level and pixel-level categories share some common tasks 3. Similar to our analysis of semantic tasks, for the 32B and 7B models, mid-level tasks benefit the most on average from finetuning, achieving the highest positive transferability in two out of three cases.

## D.4 LoRA Analysis

In this section, we analyze the cosine similarity of LoRA-finetuned weights across tasks to assess whether certain tasks induce more similar parameter updates, thereby revealing shared structure or transferable representations. For this analysis, we focus on the output projection weights from the final layer, as they exhibited the highest variance across all the layers. Figures 8, 9, and 10 show the resulting heatmaps for Qwen2.5-VL 32B, 7B, and 3B, respectively. Across all models, the strongest similarities appear among the Visual Similarity, Jigsaw, and Art Style tasks. We hypothesize that this arises because these are multi-image tasks, requiring comparable skills such as reasoning over pairs of images, assessing similarity, or aligning image composition. Consistent with the trend reported in Section 3.2, the 32B model exhibits the highest overall cosine similarity, suggesting stronger cross-task alignment in larger models. Interestingly, the 3B model shows higher similarities than the 7B model, which may be attributable to architectural differences: the 3B variant has 35 layers, whereas the 7B has 27 wider layers. A deeper interpretability analysis of these task-induced representations remains an avenue for future work.

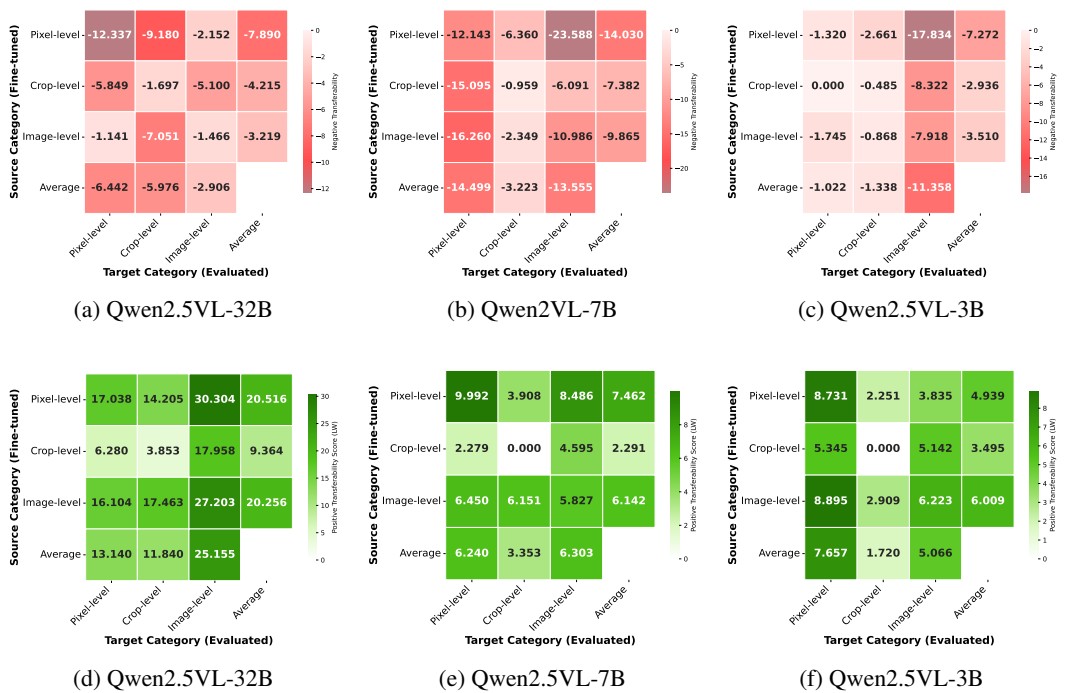

Figure 7: Comparison of transferability across semantic categories.

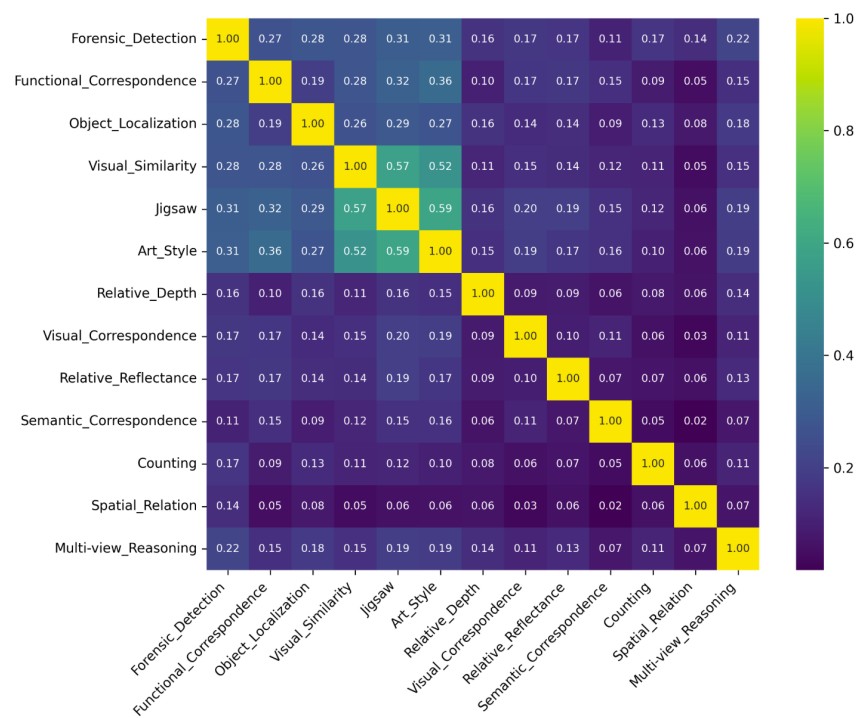

Figure 8: Cosine Similarity of LoRA weights of the output projection from layer 65 (last layer) after finetuning Qwen2.5VL-32B.

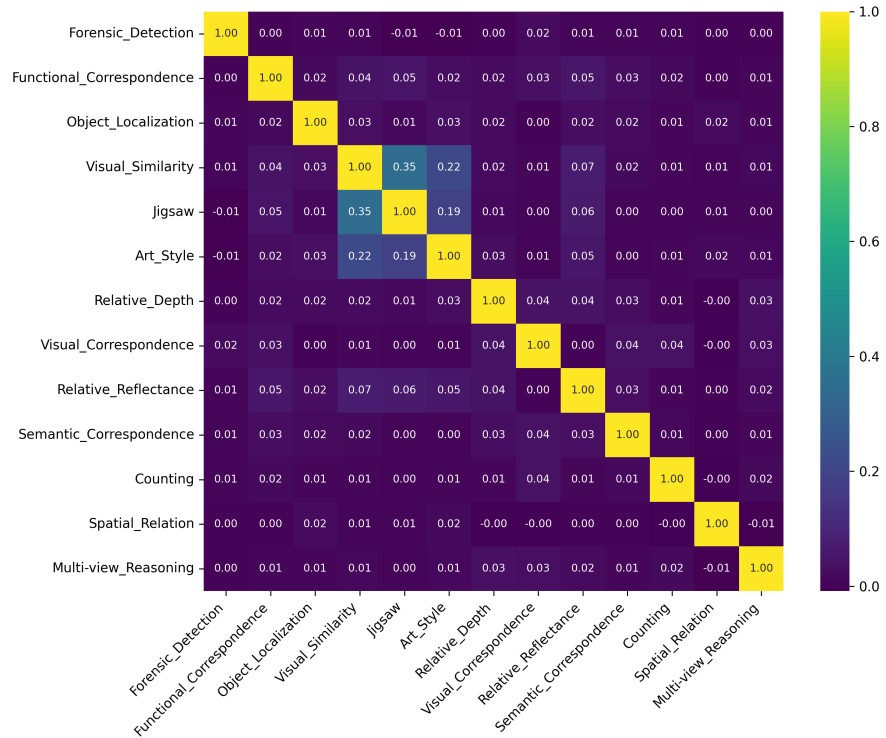

Figure 9: Cosine Similarity of LoRA weights of the output projection from layer 27 (last layer) after finetuning Qwen2.5VL-7B.

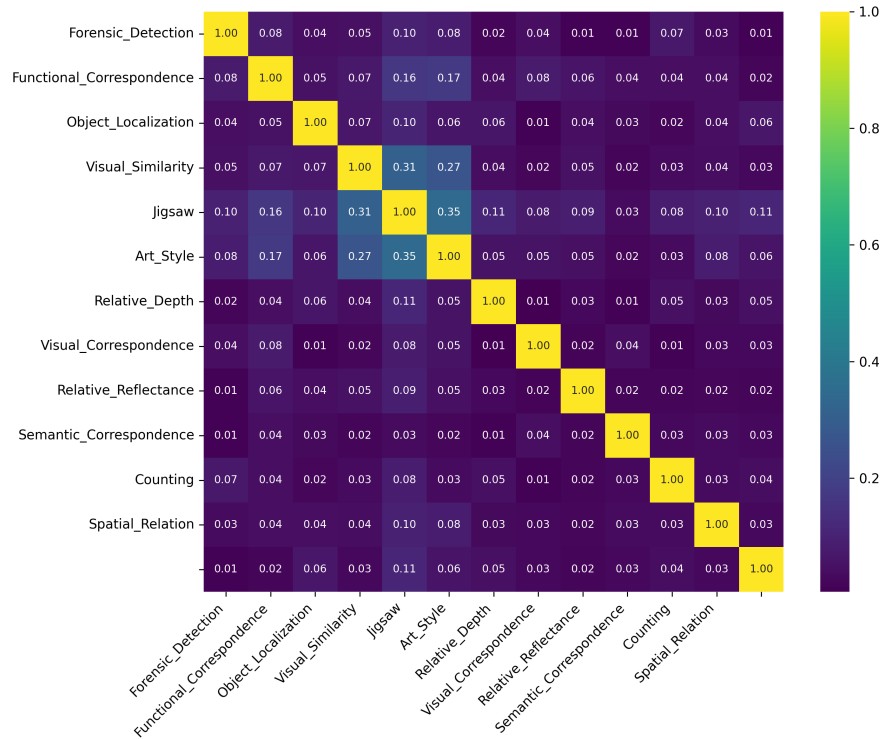

Figure 10: Cosine Similarity of LoRA weights of the output projection from layer 35 (last layer) after finetuning Qwen2.5VL-3B.

