# OpenReview forum: "Understanding Task Transfer in Vision-Language Models"
_NeurIPS.cc/2025/Workshop/UniReps — UniReps2025_

### Official Review · Reviewer_nTJm · 2025-09-05
**Review of Understanding Task Transfer in Vision-Language Models**

**Confidence:** 4

**Review:**

**Summary**
This paper presents a study of task transferability in vision-language models (VLMs), focusing specifically on visual perception tasks. The authors introduce the Performance Gap Factor (PGF) metric, intended to capture both the breadth (number of tasks affected) and the magnitude (strength of influence) of transfer effects when fine-tuning on a source task and evaluating zero-shot on others. They argue this is the first systematic study of perception task transferability in VLMs.

**Strengths**

Timely topic: Understanding transfer across perception tasks is highly relevant, as VLMs are increasingly fine-tuned on diverse objectives without clear guidance on cross-task effects.

Interesting analysis: The idea of structuring results into cliques and personas provides a useful conceptual lens for thinking about task interactions.

Metric attempt: PGF aims to normalize improvements against task performances, which is conceptually appealing when comparing across tasks with varying baselines.

Clarity of motivation: The need to analyze interference and synergy between perception tasks in multimodal models is clearly articulated.

**Weaknesses & Concerns**

Bold claims: The introduction asserts that VLMs “fall way behind human-level performance” on perception tasks. While models do underperform humans, the phrasing feels overstated given recent advances (e.g., GPT-4V and successors have narrowed gaps).

Novelty positioning: The claim of being the “first systematic study of task transferability in VLMs within the perception domain” should be tempered or better contextualized. Prior works (e.g., Taskonomy, recent multimodal transfer benchmarks) have explored related questions.

Metric clarity: The definition of PGF is somewhat confusing. In parts, “µ” is referred to as both the transferability score and as a gap measure. The authors should improve the clarity of the paper and the differences between the metric and the task transferrability. At one point, in the figure caption, PGF is refered to as Perfection instead of Performance,  the terms require correction and clearer exposition.

Ambiguity about breadth: The metric claims to capture “how many tasks are affected,” but in practice, this can only be measured for the chosen set of tasks. Generalization beyond the evaluation suite is not guaranteed, yet the paper sometimes implies a broader scope.

Overlap with prior methods: Although the authors emphasize not fine-tuning the target tasks, the overall setup is reminiscent of prior transfer learning and correlation studies, which somewhat weakens the novelty.

Minor issues:

Line 72: missing closing of absolute magnitude sign.

Terminology inconsistencies: “Performance Gap Factor” vs “Perfection Gap Factor.”

Wording in places is repetitive (principles and contributions restated multiple times).

**Overall Assessment**
The paper tackles an important and underexplored angle of VLM transferability, and the analysis is engaging. However, issues around novelty claims, clarity of metric definition, and overstatements in positioning reduce the strength of the contribution. With revisions clarifying PGF, toning down some claims, and situating the work more carefully within existing literature, the paper could provide valuable insights to the community.

**Score:**

3

**Topic Fit:**

3

---

### Official Review · Reviewer_dLjY · 2025-09-13
**Useful empirical contributions for task transfer in VLMs in the spirit of Taskonomy**

**Confidence:** 3

**Review:**

The authors introuce a "Taskonomy" variation for visual perception tasks in vision-language models, exploring the effect of task transfer on 13 perception tasks. As far as I know, they are the first to do so. They introduce a metric (PGF) to quantify transferability, compute "cliques of mutually beneficially tasks", and further demonstrate that transferability improves with model size.
To my knowledge, this work is a novel empirical contribution to the study of vision task transfer in VLMs. The methodology, i.e. the introduction of the PGF metric is sensible. The writing could be simplified and made clearer, e.g. the introduction of the terms donors, pirates, sponges and sieves was more confusing.
I suggest exploring whether the main transfer contributions stem from fine-tuning the visual encoder or the LM decoder. Furthermore, I wonder how fine-tuning on these particular tasks affects the performance on standard VQA and captioning benchmarks. For the non-archival track, this is an accept for me.

Btw: PGF is introduced as both perfection gap factor and performance gap factor. Which one is it?

**Score:**

4

**Topic Fit:**

2

---

### Official Review · Reviewer_FRN2 · 2025-09-17
**Promising study of perception-task transfer in VLMs with PGF metric and task cliques; strong discussion value, but needs clearer metric mapping, clique-mining recipe, and variance checks.**

**Confidence:** 3

**Review:**

## Summary
The submission introduces **Performance/Perfection Gap Factor (PGF)** to measure how fine-tuning a VLM on one **perception** task affects performance on other perception tasks. Using **Qwen-2.5-VL (3B/7B/32B)** across **13 BLINK tasks**, the authors visualize transfer with PGF heatmaps, discover **positive/negative task cliques**, and coin **task personas** (donors/pirates/sponges/sieves). The study targets a real gap—structured understanding of perception-task transfer in VLMs—and offers practical hooks for **curriculum design**. However, the metric normalization, clique-mining procedure, and reliability estimates need clearer specification to fully support the strongest claims.

---

## Strengths
- **Clear core idea:** PGF (and logistic-weighted transferability) provides an intuitive way to compare cross-task effects with different baselines.
- **Novelty in scope:** Focuses specifically on **perception-task transfer** (not just instruction/VQA), adding an actionable vocabulary (cliques/personas).
- **Discussion value:** Model-size trend (larger → more positive transfer) and both positive/negative cliques invite concrete workshop debate on multi-task scheduling.
- **Practical setup:** Per-task LoRA fine-tuning and zero-shot cross-task evaluation mirror realistic applied workflows.

---

## Weaknesses
- **Metric mapping under-specified:** Many tasks are not native “accuracy.” Please add a **per-task mapping table** showing how scores are normalized to [0,100], plus a **worked PGF example** for transparency.
- **Clique extraction unclear:** Edge thresholds, weighting, and the **clique-mining** method used are not specified. Please clarify the procedure and report stability across seeds/sizes.
- **Variance and reliability missing:** Results appear single-seed. Small deltas require **confidence intervals or bootstraps** (or 2–3 seeds) to make cliques/personas credible.
- **Terminology & polish:** Standardize PGF as either **Performance** or **Perfection** Gap Factor. Ensure heatmaps include **colorbars/scales** and are legible at small sizes. Deduplicate references.

---

## Questions for the Authors
- What **clique-mining algorithm** was used, and how robust are results across thresholds/seeds?
- Why define the ceiling as **100% accuracy** rather than using a task-specific ceiling (e.g., a strong baseline model)?

---

## Evaluation
The work is clear and moderately novel, with high potential to inspire discussion. The PGF metric and clique/persona framing are useful, but some technical details (metric mapping, clique-mining, variance) are under-specified. These can be fixed with light revisions or clarifications, making the abstract a strong fit.

**Score:**

3

**Topic Fit:**

3